# NLRP7 Enhances Choriocarcinoma Cell Survival and Camouflage in an Inflammasome Independent Pathway

**DOI:** 10.3390/cells12060857

**Published:** 2023-03-09

**Authors:** Déborah Reynaud, Nadia Alfaidy, Constance Collet, Nicolas Lemaitre, Frederic Sergent, Céline Miege, Emmanuelle Soleilhac, Alaa Al Assi, Padma Murthi, Gilles Courtois, Marie-Odile Fauvarque, Rima Slim, Mohamed Benharouga, Roland Abi Nahed

**Affiliations:** 1Institut National de la Santé et de la Recherche Médicale U1292, Biologie et Biotechnologie pour la Santé, 38043 Grenoble, France; 2Commissariat à l’Energie Atomique et aux Energies Alternatives (CEA), Biosciences and Biotechnology Institute of Grenoble, 38054 Grenoble, France; 3Service Obstétrique, University Grenoble Alpes and Centre Hospitalo-Universitaire Grenoble Alpes, CS 10217, CEDEX 9, 38043 Grenoble, France; 4University Grenoble Alpes, Inserm, CEA, UA13 BGE, 38000 Grenoble, France; 5Laboratory of Fundamental and Applied Bioenergetics (LBFA), Univeristy Grenoble Alpes, Inserm, 38000 Grenoble, France; 6Department of Pharmacology, Monash Biomedicine Discovery Institute, Monash University, Melbourne VIC 3800, Australia; 7Department of Obstetrics and Gynaecology, University of Melbourne, Royal Women’s Hospital, Parkville, VIC 3502, Australia; 8Departments of Human Genetics and Obstetrics and Gynecology, McGill University Health Centre Research Institute, Montréal, QC H4A 3J1, Canada

**Keywords:** NLRP7, inflammasome, gestational choriocarcinoma, camouflage, NF-κB, cancer

## Abstract

Background: Gestational choriocarcinoma (GC) is a highly malignant trophoblastic tumor that often develops from a complete hydatidiform mole (HM). *NLRP7* is the major gene responsible for recurrent HM and is involved in the innate immune response, inflammation and apoptosis. NLRP7 can function in an inflammasome-dependent or -independent pathway. Recently, we have demonstrated that *NLRP7* is highly expressed in GC tumor cells and contributes to their tumorigenesis. However, the underlying mechanisms are still unknown. Here, we investigated the mechanism by which NLRP7 controls these processes in malignant (JEG-3) and non-tumor (HTR8/SVneo) trophoblastic cells. Cell survival, dedifferentiation, camouflage, and aggressiveness were compared between normal JEG-3 cells or knockdown for *NLRP7,* JEG-3 Sh *NLRP7*. In addition, HTR8/SVneo cells overexpressing *NLRP7* were used to determine the impact of *NLRP7* overexpression on non-tumor cells. NLRP7 involvement in tumor cell growth and tolerance was further characterized in vivo using the metastatic mouse model of GC. Results: We demonstrate that NLRP7 (i) functions in an inflammasome-dependent and -independent manners in HTR8/SVneo and JEG-3 cells, respectively; (ii) differentially regulates the activity of NF-κB in tumor and non-tumor cells; (iii) increases malignant cell survival, dedifferentiation, and camouflage; and (iv) facilitates tumor cells colonization of the lungs in the preclinical model of GC. Conclusions: This study demonstrates for the first time the mechanism by which NLRP7, independently of its inflammasome machinery, contributes to GC growth and tumorigenesis. The clinical relevance of NLRP7 in this rare cancer highlights its potential therapeutic promise as a molecular target to treat resistant GC patients.

## 1. Introduction

Inflammation plays a key role during pregnancy as it controls central processes of placental development and the establishment of the feto-maternal circulation during the first trimester of pregnancy.

Gestational Trophoblastic Disease (GTD) are complications of pregnancy that encompass different disorders ranging from pre-malignant conditions represented by complete (CHM) and partial hydatidiform moles (PHM) to gestational choriocarcinoma (GC), an aggressive cancer that may develop after any pregnancy [1]. GC is a malignant, highly invasive tumor that metastasizes to multiple organs, including the vagina, the lungs, and the brain [1]. Even though PHM and CHM are usually benign diseases, patients remain at risk of developing GC. This risk is much higher for patients with CHM (20%) [2] than for patients with PHM (1.5%) [1,3]. Recent studies have demonstrated that recessive mutations in the inflammatory protein NLRP7 (Nucleotide-binding oligomerization domain, leucine-rich repeat family, Pyrin domain containing 7) are the major cause of recurrent HM [2,4,5].

NLRP7 is one of 14 NLRP proteins that are tightly associated with the control of inflammatory processes during pregnancy [6,7]. NLRP7, also known as NALP7, PYPAF3 and NOD12, is a cytoplasmic protein consisting of three main domains: (i) an N-terminal pyrin domain (PYD) involved in apoptotic and inflammatory signaling pathways, (ii) a central nucleotide-binding domain (NACHT) that promotes oligomerization and (iii) leucine-rich repeats (LRRs) domain involved in ligand sensing [8,9]. The activation of the NLRP7 inflammasome leads to the processing of Pro-Interleukin-1β to its mature form, Interleukin-1β (IL-1β), a secreted cytokine involved in the process of decidualization and trophoblast invasion [5,10].

Once activated, NLRs have been reported to mediate their pro-inflammatory signaling via two distinct pathways (i) the activation of the Caspase 1 (CASP1)-dependent inflammasome and (ii) the activation of the transcription factor Nuclear Factor-Kappa B (NF-κB) [11,12], a key activator of the transcription of Pro-IL-1β. Importantly, it was also proposed that NLRP7 might exhibit an anti-inflammatory role through its interaction with NF-κB regulatory proteins, such as the Fas-associated factor 1 (FAF-1) [13].

*NLRP7* is an NLRP member that evolved from the duplication of its paralog, *NLRP2* [14,15]. Importantly, it has been demonstrated that *NLRP2* knockdown in the trophoblastic tumor cell line, JEG-3, causes a significant increase in NF-κB p65 Ser 536 phosphorylation [16] and that this protein has the ability to inhibit the ASC-dependent activity of NF-κB (Radian et al. 2013, Kinoshita et al. 2005). Unlike *NLRP2*, *NLRP7* overexpression has been shown to inhibit Pro-CASP 1 and Pro-IL-1β production without any interference with NF-κB activity [13,17,18].

Depending on the levels of expression of NLRP7, two pathways have been proposed for its function [3]. While normal levels of NLRP7 have been associated with inflammasome activity, its overexpression has been proposed to function in an inflammasome-independent manner [19]. A recent study from our group substantiated this observation by demonstrating that NLRP7 is overexpressed in trophoblast tumor cells, the JEG-3 cells, which do not express IL-1β [2]. These findings strongly suggested that the NLRP7 functions in an inflammasome-independent manner in trophoblastic tumor cells. Recent studies also demonstrated that NLRP7 is involved in embryonic development and in the regulation of the humoral immune response [20,21] and that its overexpression drives the differentiation of decidual macrophages from the pro-inflammatory type M1 to the anti-inflammatory type M2 [22].

Altogether, these findings strongly suggest that NLRP7, through these functions, may contribute to the evasion of tumor cells from the maternal immune system and their camouflage, which in turn may contribute to their growth and invasiveness. However, the mechanisms by which this occurs have not been elucidated.

The objectives of this study were (i) to determine the underlying mechanisms of NLRP7 function in JEG-3 cells; (ii) to characterize NLRP7 functions, in an inflammasome-independent manner, on JEG-3 cells survival, camouflage and dedifferentiation; (iii) to identify the NLRP7-mediated pathway that drives JEG-3 cells into tumorigenesis; and (iv) to validate these results in vivo, using a metastatic mouse model of GC injected with JEG-3 Sh CTL and JEG-3 Sh *NLRP7* cells.

## 2. Materials and Methods

### 2.1. Cell Culture

#### 2.1.1. Culture of HTR8/SVneo

The cell line HTR8/SVneo (ATCC^®^ CRL3271™) is a normal human extravillous trophoblast cell line that was created upon transfection of cells derived from human first-trimester placenta, with the gene that encodes simian virus 40 large T antigen. The HTR8/SVneo cell line was cultured in RPMI 1640 medium that was supplemented with 5% FBS, penicillin-streptomycin, and amphotericin B (Invitrogen, Cergy Pontoise, France). The cells were maintained in a humidified incubator at 37 °C with 5% CO_2_. In the figures, the HTR8/SVneo cell line is referred to as HTR.

#### 2.1.2. Culture of JEG-3 Cell Line

The human choriocarcinoma cell line JEG-3 (ATCC^®^ HTB-36 TM) is one of six clonally derived cell lines isolated from the Woods strain of the Erwin-Turner tumor by Kohler and associates [23]. The JEG-3 cell line was grown in DMEM/F-12+GlutaMAX medium supplemented with 10% FBS, penicillin-streptomycin, and gentamicin (Invitrogen, Cergy Pontoise, France). Cells were maintained in a humidified incubator at 37 °C with 5% CO_2_.

#### 2.1.3. JEG-3 Luciferase and HTR8/SVneo Luciferase Preparation

JEG-3-Luc and HTR8/SVneo-Luc cells expressing Luciferase were generated by transducing lentivirus pLenti-II-CMV-Luc-IRES-GFP-neomycin into the JEG-3 cell line and pLenti-II-CMV-Luc-IRES-GFP-blasticidin into the HTR8/SVneo cell line, as per the manufacturer’s instructions (Applied Biological Materials Inc, Viking Way, Canada.). The cells were transduced with lentivirus for 4 h in a fresh culture medium containing Polybrene (8 µg/mL, Sigma Aldrich, Henri Desbruères, France). The transduced cells were then selected for 10 days using Neomycin (200 μg/mL) for JEG-3 and Blasticidin (10 µg/mL) for HTR8/SVneo.

#### 2.1.4. JEG-3 Luciferase Sh NLRP7 and HTR8/SVneo-Luciferase Sh NLRP7 Preparation

The lentivirus supernatant used for cell preparation was derived from bacteria that were transformed with Sh NLRP7 plasmids and Sh Control (MISSION pLKO.1-puro; Sigma Aldrich). The protocol followed the instructions provided by the company (Sigma Aldrich). Cells were transduced with the lentivirus for 4 h at a 1:2 ratio in a complete medium containing Polybrene (8 µg/mL, Sigma Aldrich). Infected cells were selected with puromycin at 10 µg/mL for 14 days, and drug-resistant cells were collected after 2–3 weeks for single-cell cloning in 96-well plates. Wells with more than one cell were excluded from further investigation. Fifty percent of the medium in each well was replaced twice a week. Surviving clones reached confluency after 4–6 weeks and were expanded for banking. The resistant clones were confirmed to be invalidated for NLRP7 by western blot and RT-qPCR. The steadily transduced cells were maintained in puromycin at a final concentration of 2 µg/mL.

#### 2.1.5. Overexpression of NLRP7 in HTR8/SVneo Cell Line

HTR8/SVneo cells were transfected using Lipofectamine 3000 (ThermoFisher Scientific, Life Technologies, Villebon-sur-Yvette, France) with pcDNA3.1 ± NLRP7-c-myc (6 µg). The plasmid was kindly provided by Dr. C. Stehlic (Division of Rheumatology, Department of Medicine, Feinberg School of Medicine, Northwestern University, Chicago, IL 60611, USA). C-myc protein was used as a tag to validate NLRP7 overexpression in HTR8/SVneo cells.

#### 2.1.6. Cell Treatments

Cells were treated with TNF-α (10 ng/mL) (Sigma Aldrich, Henri Desbruères, France), an activator of the NF-κB pathway, for up to 30 min at 37 °C, 5% CO_2_. In another way, cells were treated with FSL-1, a bacterial-derived toll-like receptor 2/6 agonist at 0.1 µg/mL (Invivogen, Toulouse, France), a specific activator of NLRP7.

#### 2.1.7. Transfection of siRNA and Dual-Luciferase Reporter Gene Assay in JEG-3 Cells

JEG-3 and HTR8/SVneo cells were seeded in 48-well plates (60,000 cells per well for JEG-3 and 40,000 cells for HTR8/SVneo) with their respective media. 24 hrs post-seeding, both cell types were transfected with small interfering RiboNucleic Acid (siRNA) for *NLRP7* (Ambion^®^ 2.5 pM, *n* = 4) and for negative control (Ambion^®^ 2.5 pmol, *n* = 4) with Lipofectamine RNAiMax (ThermoFisher Scientific, Life Technologies) in OptiMEM media (Gibco, Thermo Fisher, Illkirch, France) for 24 h. To measure NF-κB activity, cells were co-transfected, using Lipofectamine 3000 (ThermoFisher Scientific, Life Technologies, Villebon-sur-Yvette, France) with the luciferase reporter plasmid κB-Luc (2.5 µg) that contains an NF-κB responsive element and with the plasmid TK-Renilla (25 ng) containing the Renilla luciferase gene. The latter was used as an internal control to estimate the transfection efficiency. The plasmids were kindly provided by Dr. Gilles Courtois [24]. To activate the NF-κB pathway, cells were treated with TNF-α (10 ng/mL) for up to 30 min during the transfection. Firefly and Renilla luciferase activities were measured using the Twinlite Dual Luciferase Reporter Gene Assay System (PerkinElmer, Villebon-sur-Yvette, France) on a Spark^®^ reader (Tecan, Lyon, France). Results are presented as relative luminescence units of Firefly luciferase activity over luminescence units of Renilla luciferase activity.

### 2.2. Cell Death Assay (IncuCyte, EssenBioScience, Inc.)

JEG-3 Sh CTL and JEG-3 Sh NLRP7 cells were seeded at 10,000 cells/well in a 96-well plate without FBS. They were treated with 5 µg/mL propidium iodide (PI) and allowed to grow for 48 h. An IncuCyte Zoom System was used to track the cells, and bright field images were taken every 2 h throughout the experiment. The Zoom^®^ software was used to analyze the rate of cell death. The ratio of red fluorescent (PI-positive cells) to total cell confluence was plotted against the treatment time (*n* = 12).

### 2.3. ELISA Test

The IL-1β ELISA kit for measuring inflammatory cytokines was obtained from Pepro-Tech (Pepro-Tech, Neuilly-sur-Seine, France). The cytokine levels were measured in the collected sera of mice. The ELISA tests were conducted according to the manufacturer’s protocols, and all samples fell within the linear range of the standard curves.

### 2.4. Lactate Measurement Assay

To quantify the concentration of lactate in biological samples, JEG-3 cell culture supernatants were incubated for 90 min in a 96-well plate with the following buffer solution: NAD+ (Roche Diagnostics; 0.75 mM), Hydrazine monohydrate (Sigma Aldrich, Henri Desbruères, France; 0.4 M), Glycine (EUROMEDEX, Souffelweyersheim, France; 0.4 M) and L-LDH enzyme (Roche, Meylan, France; 40 Units/well). In each well, lactate was converted back into pyruvate via the reduction of NAD+ into NADH and H+ (stoichiometry 1:1) by L-LDH. The absorbance of NADH was measured at 340 nm using a microplate reader before and after the addition of the enzyme L-LDH (40 units/well). Lactate concentration was calculated from NADH absorbance value using the Beer–Lambert law, considering the dilution factor and stoichiometry.

### 2.5. LDH Activity Assay

When cell membranes are compromised or damaged, the enzyme lactate dehydrogenase (L-LDH) is released into the surrounding extracellular space. The presence of this enzyme in the culture medium can be used as a cell death marker. To assess the release of L-LDH from biological samples, JEG-3 cell culture supernatants were incubated with the following buffer solution: NAD^+^ (Roche, Meylan, France; 0.75 mM), Lactic acid sodium salt (Sigma Aldrich, Henri Desbruères, France; 1 mM), Hydrazine monohydrate (Sigma; 0.4 M) and Glycine (EUROMEDEX; 0.4 M). LDH activity in samples was determined kinetically by monitoring NADH absorbance at 340 nm for 90 min using a microplate reader.

### 2.6. Protein Extraction and Immunoblotting

The cells were homogenized using RIPA (RadioImmunoPrecipitation Assay) lysis buffer containing a protease inhibitor cocktail (Sigma Aldrich). The lysis buffer contained Tris-HCl (2.42 g), NaCl (8.77 g), sodium deoxycholate (5 g), sodium dodecyl sulfate (SDS) (1 g), Triton X-100 (10 mL), and had a pH of 8. The homogenates were centrifuged at 11,000× *g* for 30 min at 4 °C, and the resulting supernatants were collected. The protein concentrations of the supernatants were determined using a Spark^®^ reader (Tecan) and the micro-BCA protein assay. The extracted proteins were mixed with 5X protein sample buffer containing Tris-HCl (62 mM, pH 6.8), SDS (2%), glycerol (10%), β-mercaptoethanol (5%), and bromophenol blue (0.05%) as the tracking dye. The samples were then heated at 95 °C for 5 min.

#### Western Blot Analysis

Samples were loaded onto 4–20% SDS-PAGE gels and transferred to nitrocellulose membranes (0.22 µm, Bio-Rad, Marnes-la-Coquette, France) using the Trans-Blot Turbo Transfer System (Bio-Rad). The membranes were blocked in 5% non-fat dry milk-TBS Tween 0.05% and incubated overnight at 4 °C with primary antibodies, including in-house polyclonal anti-NLRP7 (used at 3.5 µg/mL, Covalab, Lyon, France), monoclonal anti-IL-1β (used at 1 µg/mL, Santa Cruz, Heidelberg, Germany), monoclonal anti-OCT3/4 (used at 1 µg/mL, Santa Cruz), monoclonal anti-NF-κB (used at 1 µg/mL, Santa Cruz), monoclonal anti-HLA-G (used at 1 µg/mL, Covalab, Lyon, France), monoclonal anti-PDL-1 (used at 1 µg/mL, Cell Signaling, Saint-Cyr-L’École, France), monoclonal anti-phospho-IκBα (used at 1 µg/mL, Cell Signaling, Saint-Cyr-L’École, France), and monoclonal anti-IκBα (used at 1 µg/mL, Cell Signaling). The membranes were then incubated for 1 h with corresponding horseradish peroxidase-labeled secondary antibodies (1:10,000). Chemiluminescence detection kit reagents and a ChemiDoc imaging system (Bio-Rad, Marnes-la-Coquette, France) were used to detect immunoreactivity. To normalize for sample loading, the blots were subsequently stripped using a commercially available kit (Re-blot, Millipore, Molsheim, France) and re-probed with anti-β-Actin antibody ((Sigma Aldrich, Henri Desbruères, France) or anti-Tubulin ((Sigma Aldrich, Henri Desbruères, France) as an internal control.

### 2.7. Quantification of NF-κB Nuclear Translocation Using High Content Imaging and Analysis (HCA)

Cells were plated in a black clear bottom 96-well plate at 10,000 per well. To activate the NF-κB pathway, TNF-α at 10 ng/mL was added and incubated for 30 min. Control cells were treated with a TNF-α vehicle as a negative control. The cells were washed and fixed with 4% paraformaldehyde (PFA) for 15 min, followed by permeabilization with 0.5% Triton X-100 for 5 min. To block non-specific binding, PBS 1X 5% horse serum was added and incubated for 1 h. The cells were then labeled with a mouse anti-NF-κB primary antibody (Santa Cruz, Heidelberg, Germany) and a donkey anti-mouse Alexa 488 secondary antibody (Jackson ImmunoResearch Laboratories, Montlucon, France) for 1 h each. Nuclei were labeled with Hoechst 33258 (10 µg/mL, Sigma-Aldrich).

Eight images per well were captured using an automated microscope ArrayScanVTI (Thermo Scientific, Villebon-sur-Yvette, France) with a Zeiss 10x (NA 0.4) LD Plan-Neofluor. The dichroic mirrors used for Hoechst and NF-κB were BGRFR-386/23 and BGRFR-585/205, respectively. The exposure times were set using control wells (with or without TNF-α) to reach 25% and 40% of Hoechst and NF-κB fluorescence intensity, respectively. The Molecular Translocation Bio-Application of Thermo Scientific HCS Studio v 6.5.0 was used for quantification of the translocation of NF-κB from the cytoplasm to the nucleus.

Each nucleus was detected in the Hoechst channel using the isodata thresholding method. Border-touching nuclei were rejected from each image. The Circ mask in the NF-κB channel was created using individual nuclei identified in the Hoechst channel. A Ring region was defined in the cytoplasm beyond the nuclear (i.e., Circ mask) region. The Circ and Ring masks were applied only to objects that passed the object selection criteria based on a mean pixel intensity set at 1200 arbitrary units (a.u.) in the NF-κB channel. The average intensity was measured within the Circ and Ring masks, allowing reporting of both the difference and ratio between the average pixel intensity measured in the Circ and Ring masks for the NF-κB channel. The arbitrary units correspond to the raw fluorescence intensities obtained through the HCS Studio software. Results were presented as the percentage of fluorescence of NF-κB in the nuclei.

### 2.8. Total RNA Isolation and RT-qPCR Analyses

The Macherey Nagel RNA extraction kit was used to extract total RNA from HTR8/SVneo and JEG-3 cells following the manufacturer’s protocol. Reverse transcription was conducted on 1 µg of total RNA using the Iscript kit from Biorad. Table 1 lists the primers used in the study. RT-qPCR was performed using the comparative threshold (CT) method based on the ΔΔCt approach after determining the CT values for both the reference and target genes in each sample. SYBR Green Supermix and the Bio-Rad CFX96 apparatus (Bio-Rad) were used. The housekeeping gene 18SrRNA was used to determine the relative mRNA levels.

### 2.9. Total RNA Sequencing Analysis

The Illumina TruSeq Stranded mRNA Library Prep Kit was used to prepare RNA sequencing (RNA-seq) libraries with 150 ng of input RNA. Paired-end RNA-seq was carried out on an Illumina NextSeq sequencing instrument (Helixio, Clermont-Ferrand, France). The RNA-seq read pairs were then mapped to the reference human GRCh38 genome using Bowtie and TopHat. Statistical analyses were performed using R software (version 3.2.3, Free Software Foundation, Inc. 51 Franklin St, Fifth Floor, Boston, MA, USA) and R packages developed by CummeRBund. The expression levels of each gene were summarized and normalized using Cuffdiff. Differential expression analysis was performed using the FPKM pipeline, with *p*-values adjusted to control the global FDR across all comparisons with the default option of the FPKM package. Genes were considered differentially expressed if they had an adjusted *p*-value less than 0.05 and a fold change of at least 1.2. Pathway enrichment analyses were conducted using the Gene Set Enrichment Analysis software with an online curated gene set collection available at http://www.genome.jp/kegg/ (accessed on 28 January 2022).

### 2.10. Animal Model Study

#### 2.10.1. Experimental Groups

The in vivo experiments were conducted in accordance with the guidelines set by the European Community for the Use of Experimental Animals and were approved by the institutional ethics committee. Female SHO SCID mice, aged between two to three months, were mated and housed in the animal facility. Female mice were housed in groups of 4 mice per cage, which favors menstrual synchronization [25]. Mice were enrolled blindly in two groups. Mice enrolled in group 1 (*n* = 7) were injected with JEG-3 Sh CTL, while mice of group 2 (*n* = 7) were injected with JEG-3 Sh NLRP7 via their tail vein.

#### 2.10.2. Bioluminescence Imaging

Mice were imaged for up to 34 days after injection. Bioluminescence imaging was performed using a highly sensitive, cooled CCD camera mounted in a light-tight specimen box (IVIS^®^ In Vivo Imaging System; PerkinElmer). Animals were anesthetized with 2% isoflurane before imaging. To prepare for imaging, the mice were injected with 10 µL/10 g of body weight of luciferin (potassium salt, Xenogen, Alameda, CA, USA) 15 min prior to imaging. This dose and route of administration have been reported to be optimal for studies conducted in rodents. Images were acquired within 15 min post-luciferin administration. [26]. To perform the imaging, the mice were positioned on a heated stage within a light-sealed camera box and were continuously exposed to 1–2% isoflurane. Bioluminescence was detected with the IVIS^®^ camera system for 45 s, which has been shown to yield optimal results. The low levels of light emitted from the tumors were captured, digitized, and displayed. The tumor areas were identified and quantified using Living Image^®^ software as total photons per second. Following whole-body imaging, a laparotomy was conducted to collect blood and visualize potentially metastatic organs. The metastatic organs were either stored at −80 °C or collected in 4% PFA for immunohistochemical analyses.

#### 2.10.3. Histology and Immunohistochemistry of Mouse Tissues

Tissues were processed as described previously [2,27]. The lung tissues were subjected to immunohistochemistry using the following antibodies: anti-Ki67, clone MIB1 antibody at a concentration of 0.61 µg/mL (Dako, Carpinteria, CA, USA catalog number M7240); anti-hCG, ready-to-use antibody (Dako, Carpinteria, CA, USA catalog number IS50830-2); anti-PD-L1 (Cell signaling, catalog number 13684); anti-HLA-G clone 4H84 (Covalab, lyon, France catalog number MAB20300); and anti-F4/80 clone BM8 (eBioscience, Villebon-sur-Yvette, France catalog number 14-4801-82).

#### 2.10.4. Cytokine Arrays

The levels and secretion profiles of cytokine-related proteins in the sera of both mouse groups were compared using an antibody array. The Mouse Cytokine Antibody Array kit (#1333995 Abcam, Cambridge, UK) was used in accordance with the manufacturer’s instructions. The immunoreactive band intensities were measured by scanning the photographic film and analyzing the images using ImageJ software and the Chemidoc analyzing system (Image Lab version 4.0.1).

### 2.11. Statistical Analysis

Data were analyzed using Mann–Whitney, Student’s *t*-test, and 1-way ANOVA statistical tests. Normality and equal variance were checked before conducting the tests. In cases where normality failed, a nonparametric test followed by Dunn’s or Bonferroni’s test was used. SigmaPlot, SigmaStat, Jandel Scientific Software (version 12.5 from Systat Software, Inc., San Jose, CA, USA), and GraphPad Prism 8 (version 8, GraphPad Software, Franklin Street. Fl. 26, Boston, MA, USA) were used for statistical analysis. The results are presented as means ± SEM with statistical significance levels of *p* < 0.001, 0.01, and 0.05.

## 3. Results

### 3.1. NLRP7 Functions in an Inflammasome-Independent Manner in JEG-3 Cells

We previously demonstrated that *NLRP7* is highly expressed in the human GC cell line, JEG-3, compared to the normal (non-tumor) trophoblast cell line, HTR8/SVneo. We also demonstrated that JEG-3 cells do not express IL-1β, the main effector of NLRP7 inflammasome [2]. To investigate why JEG-3 cells do not express IL-1β, we activated NLRP7 inflammasome by FSL-1, both in JEG-3 and HTR8/SVneo cells. Figure 1A shows that *IL-1β* mRNA was only detected in HTR8/SVneo but not in JEG-3 cells. The stimulation of HTR8/SVneo cells with FSL-1 did not affect *IL-1β* mRNA level but significantly increased its maturation and secretion, whereas no changes were observed in JEG-3 cells upon FSL-1 stimulation, Figure 1B,C. To characterize the link between the levels of expression of *NLRP7* and those of *IL-1β*, we compared the levels of *IL-1β* mRNA in HTR8/SVneo and JEG-3 cells after knocking down the *NLRP7* gene (Sh strategy). Figure 1D shows a significant decrease in the mRNA levels of *IL-1β* in HTR Sh *NLRP7* and a total extinction of IL-1β secretion as compared to HTR Sh CTL, Figure 1E. Knockdown of *NLRP7* in JEG-3 cells did not change the levels of *IL-1β* mRNA or its secretion, which were not produced by these cells before knocking-down *NLRP7* RNA sequencing (RNA-seq) of JEG-3 Sh CTL and JEG-3 Sh *NLRP7* cells revealed that both cells do not express three key genes of the NLRP7 inflammasome machinery, *ASC*, *CASP 1* and *IL-1β*, Appendix A. The lack of *CASP 1* expression was also confirmed by RT-qPCR, Appendix A. RNA-seq analysis also identified an increase in the expression of the closest NLRP gene to *NLRP7*, i.e., the *NLRP2* gene, Appendix A. Using RT-qPCR, we further demonstrated that JEG-3 expresses significantly more *NLRP2* compared to HTR8/SVneo, Appendix A and that JEG-3 Sh *NLRP7* expresses 2.5-times more *NLRP2* than JEG-3 Sh CTL, Appendix A. These findings demonstrate that similar to *NLRP7*, *NLRP2* expression is increased in JEG-3 cells. Altogether, these findings suggest that NLRP7 overexpression in JEG-3 cells not only impacted the expression of its inflammasome machinery but may also impact the expression of other NLRP family members.

### 3.2. NF-κB Pathway Is Downregulated in JEG-3 Choriocarcinoma Cells

Because the NF-κB pathway contributes to Pro-IL-1β production [28,29], we first compared the levels of NF-κB protein expression in JEG-3 versus HTR8/SVneo cells. Figure 2A,B show that NF-κB protein is expressed at much lower levels in JEG-3 cells compared to HTR8/SVneo cells. We then compared NF-κB activation in HTR8/SVneo and JEG-3 cells through the assessment of the IκBα phosphorylation. Cells were treated with 10 ng/mL of TNF-α for different time points (0, 10, and 30 min) and compared for IκBα protein phosphorylation, Figure 2C,D. We demonstrated that after 10 min of stimulation, the NF-κB pathway was activated more strongly in HTR8/SVneo cells than in JEG-3 cells.

### 3.3. NLRP7 Activates NF-κB Pathway More Strongly in Non-Tumor Trophoblastic Cells

The above experiments strongly suggested that the NF-κB pathway is less active in JEG-3 cells and raised the question as to whether high NLRP7 expression in these cells contributes to the decreased activation of NF-κB. To verify this hypothesis, we compared NF-κB activation in HTR8/SVneo and JEG-3 cells that were transfected by a luciferase reporter plasmid κB-Luc containing an NF-κB responsive element. These cells were beforehand invalidated or not for *NLRP7* using the siRNA strategy. All cell types were stimulated or not with TNF-α. Figure 2E,F shows that TNF-α significantly increased NF-κB activation in both cell types, albeit to a variable extent. NF-κB activation in response to TNF-α was more than 10 times (see *Y* axes) stronger in HTR8/SVneo cells (Figure 2E) than in JEG-3 cells (Figure 2F). *NLRP7* knockdown did not significantly affect the activation of NF-κB in JEG-3 cells (Figure 2F) but significantly decreased the activation of NF-κB in HTR8/SVneo cells (Figure 2E). These data demonstrate that the NF-κB pathway is less activated in JEG-3 tumor cells and that NLRP7 contributes to NF-κB pathway activation in non-tumor HTR8/SVneo cells.

To determine whether NLRP7 contributes to NF-κB translocation into the nucleus, the NF-κB immunofluorescence labeling strategy was conducted using HTR Sh CTL and Sh *NLRP7* and JEG-3 Sh CTL and Sh *NLRP7* under basal or stimulated conditions by TNF-α. Cellular translocation of NF-κB into the nucleus was detected and quantified in all conditions, Figure 2G. Following activation by TNF-α, a significant increase in the translocation of NF-κB into the nucleus was observed in all conditions, HTR Sh CTL/Sh *NLRP7* and JEG-3 Sh CTL/Sh *NLRP7*. We observed a significant decrease in the translocation of NF-κB into the nucleus upon *NLRP7* knockdown. This decrease was more important in HTR8/SVneo than in JEG-3 cells and was more pronounced when cells were treated with TNF-α. Altogether, these results demonstrate that NLRP7-mediated NF-κB activation is mainly operating in non-tumor cells, which may be related to the lack of expression of NLRP7 inflammasome effectors in JEG-3 cells.

### 3.4. NLRP7 Promotes the Survival of JEG-3 Cells and Contributes to Their Aggressiveness

We previously demonstrated that NLRP7 promotes choriocarcinoma growth and that its knockdown suppresses cellular proliferation [2]. Herein, we evaluated the impact of *NLRP7* knockdown on JEG-3 cell survival using three strategies, the monitoring of propidium iodide fluorescence, assessment of LDH activity in culture medium and quantification of cellular lactate levels. Figure 3A shows a comparison of the percentage of cell death between JEG-3 Sh CTL and JEG-3 Sh *NLRP7* following 60 h of culture without FBS. We demonstrate that knocking down *NLRP7* in JEG-3 significantly decreased JEG-3 survival. In control cells, 50% of cells die after 40 h of culture, whereas in JEG-3 Sh *NLRP7,* this percentage is reached much earlier, after 18 h of culture.

Since aerobic glycolysis contributes to tumor cells’ aggressiveness and is characterized by a massive increase in glucose consumption and lactate production [30], we compared lactate concentration in the conditioned media of JEG-3 Sh CTL and JEG-3 Sh *NLRP7* cells at different time points. Figure 3B shows a significant decrease in lactate levels in JEG-3 Sh *NLRP7* compared to JEG-3 Sh CTL at 24 and 48 h of incubation. This result substantiates the direct involvement of NLRP7 in the acquisition of aggressiveness by JEG-3 cells. We then compared the lactate dehydrogenase activity (LDH) in the supernatants of the same cells. An increase in LDH activity in the cellular supernatants indicates that plasma membranes are affected, and this correlates with cell death. We observed a significant increase in LDH activity in the culture media collected from JEG-3 Sh *NLRP7* at both 24 and 48 h, Figure 3C. The increased release of LDH from JEG-3 Sh *NLRP7* cells indicates that *NLRP7* knockdown decreases JEG-3 cell survival.

NLRP7 implication in cell survival was further investigated in JEG-3 Sh CTL and JEG-3 Sh *NLRP7* cells by comparing the mRNA levels of three key genes known to be involved in cell survival. These include the ligand MIF, its receptor CD74 and its co-receptor CXCR2. Figure 3D and Appendix A show that the levels of *MIF*, *CD74* and *CXCR2* mRNA were significantly decreased in JEG-3 Sh *NLRP7* compared to JEG-3 Sh CTL cells supporting further the involvement of NLRP7 in JEG-3 cell survival and aggressiveness.

### 3.5. NLRP7 Increases JEG-3 Dedifferentiation

We and others have demonstrated that NLRP7 is involved in non-tumor cell differentiation [5,31]. However, no data were available regarding the potential involvement of NLRP7 in the dedifferentiation of JEG-3 cells. Here, we compared the mRNA levels in JEG-3 Sh CTL and JEG-3 Sh NLRP7 cells of three key genes, OCT3/4, NANOG, and NOTCH1, known to be involved in cell differentiation. We found that the levels of these three genes were significantly decreased in JEG-3 Sh NLRP7, Figure 4A–C. The decrease in OCT3/4 protein was also confirmed by western blot analysis, Appendix A. These data strongly suggest that high NLRP7 expression in JEG-3 cells may enhance or maintain their dedifferentiated state, which could trigger or contribute to their aggressiveness. To further investigate the impact of elevated NLRP7 expression on JEG-3 cell dedifferentiation, we used RNA-seq analysis to compare changes in gene expression and pathways between JEG-3 Sh CTL and JEG-3 Sh NLRP7. Differential gene expression analysis using the DEseq2 method identified 160 genes that were significantly differentially expressed (adjusted *p*-value < 0.05 and absolute log2 fold change > 1.5), with 46 upregulated (represented by red dots) and 114 downregulated (represented by blue dots) in JEG-3 Sh NLRP7. A volcano plot was generated to visualize the 160 differentially expressed genes, with the logarithm of the *p*-value plotted against the logarithm of the fold change of expression between JEG-3 Sh NLRP7 and JEG-3 Sh CTL (Figure 4D). Further analysis using gene set enrichment analysis (GSEA) revealed that decreased NLRP7 expression suppressed responses to macrophage colony-stimulating factor (MCSF), increased the humoral immune response, and activated processes leading to the acquisition of embryonic markers, Figure 4E–G. These findings provide additional evidence of the involvement of high NLRP7 expression in the maintenance of tumor cells in an undifferentiated state.

### 3.6. NLRP7 Contributes to the Camouflage of GC Cells

Recent data from our group [2] and the above RNA-seq analyses strongly suggested that NLRP7 is involved in the camouflage of JEG-3 cells from the maternal immune system. However, the mechanism by which this occurs has not been elucidated. Among known genes involved in trophoblast camouflage are *PD-L1*, *HLA-G*, *HLA-C* and *HLA-E* [16,32,33,34,35,36,37,38]. RNA-seq analyses showed a decrease in the levels of expression of *PD-L1*, *HLA-G* and *HLA-C* after *NLRP7* invalidation, Appendix A.

These changes were confirmed by RT-qPCR for *PD-L1*, *HLA-G*, *HLA-C* and *HLA-E*, Figure 5A–D, and by Western blot analyses for PD-L1 and HLA-G, Figure 5E–H. To confirm the direct involvement of NLRP7 in the regulation of these genes, we overexpressed NLRP7 in HTR8/SVneo (OE) and compared their levels of expression between HTR-OE and HTR CTL. Appendix A shows that the overexpression of NLRP7 caused significant increases in HLA-G, OCT 3/4 and PD-L1. Altogether, these data demonstrate the direct involvement of NLRP7 in the regulation of these genes and substantiate its contribution to the processes of camouflage and dedifferentiation of JEG-3 cells.

### 3.7. NLRP7 Is Involved in GC Tumor Growth and Camouflage in the Lungs

To validate the data obtained in vitro in an integrated mouse model of GC, we injected JEG-3 Sh CTL or JEG-3 Sh *NLRP7* into the tail vein of female mice. We have previously shown that this method allows tumor cells to rapidly disseminate throughout the body via blood circulation and to a niche in the typical metastatic organ of GC, the lungs. Figure 6A shows that invalidation of *NLRP7* in JEG-3 cells significantly compromised tumor development in the lungs. Quantification of tumor growth showed that a significant difference was observed from day 28 post-injection. A trend of an increase was also observed at day 34 post-injection, Figure 6B.

### 3.8. JEG-3 Sh NLRP7 Cells Are Less Tolerated by the Immune System

We used antibody array analysis to compare a selection of circulating inflammatory cytokines known to be involved in the immune response. We performed this assay on blood collected from mice injected with JEG-3 Sh CTL or JEG-3 Sh *NLRP7* cells.

Figure 7A shows significant increases in the levels of TECK, PF4, IL-1β, and TNF-α cytokines and significant decreases in ILR3R, MCSF, TIMP1, SDF1a and IL6 cytokines in the JEG-3 Sh *NLRP7* group of mice. Importantly, most of the upregulated cytokines were reported to be associated with immune cell priming to induce anti-tumor responses in the tumor microenvironment and were also reported to be involved in innate and adaptive immunity [39,40,41,42]. In contrast, most of the downregulated cytokines were reported to be linked to immunosuppressive functions [39,43,44,45,46,47,48]. Because IL-1β is a master cytokine that controls local inflammatory responses, we used a specific ELISA test to confirm its increase in the circulation of the JEG-3 Sh *NLRP7* group of mice, Figure 7B. These data strongly suggested that *NLRP7* knockdown in JEG-3 cells may contribute to the reactivation of the mouse immune response.

### 3.9. JEG-3 Sh NLRP7 Tumors Are Less Proliferative and More Visible to the Immune System

Histological comparison of the lungs collected from mice injected with JEG-3 Sh CTL or JEG-3 Sh NLRP7 showed that tumor development was observed in 71% of lungs collected from the JEG-3 Sh CTL group versus only 14% of lungs from the JEG-3 Sh *NLRP7* group. Tumors in the JEG-3 Sh *NLRP7* group exhibited smaller and more necrotic tumor masses, Figure 7C(a,b). To better characterize these lung tumors, we compared their immunoreactivity to Ki67, hCG, HLA-G, PD-L1, and F4/80, a mouse protein marker of macrophages. Figure 7C(c–n) report representative photographs of the immunoreactive protein expression in these cells. A weak decrease in proliferating (c,d) and hCG-secreting (e,f) cells was observed in JEG-3 Sh *NLRP7* cells. These cells also showed less immunoreactivity for HLA-G (g,h) and PD-L1 (i,j). F4/80 (k,l) appeared to be contiguous to the JEG-3 Sh CTL and JEG-3 Sh *NLRP7* tumors with no observed infiltration.

### 3.10. Proposed Model of NLRP7 Mechanism of Function in Normal and GC Cells

This study brought strong evidence for the direct involvement of high NLRP7 expression in GC growth and aggressiveness. The graphical summary in Figure 8 illustrates the mode of function of NLRP7 in non-tumor and tumor trophoblast cells. In non-tumor trophoblast cells, NLRP7 is expressed at normal levels and functions in an inflammasome-dependent manner. In these cells, NLRP7 activates the NF-κB pathway and induces its translocation to the nucleus, which in turn induces the transcription of Pro-IL-1β that matures into IL-1β. NLRP7 also regulates the expression of HLA-G, HLA-E, HLA-C and PD-L1, contributing to trophoblast tolerance by the maternal immune system, and favors macrophage polarization to the M1 subtype creating a pro-inflammatory environment. All these events contribute to the protection of the fetus and ensure the safe progression of the pregnancy.

In tumor trophoblast cells, NLRP7 is overexpressed and functions in an inflammasome-independent manner. It inhibits IL-1β production by decreasing NF-kB activation. NLRP7 also mediates the overexpression of HLA-G, HLA-E, HLA-C, and PD-L1 and favors macrophage polarization to the M2 subtype. These events increase the maternal immune tolerance of tumor cells, which creates a favorable anti-inflammatory environment that contributes to tumor growth. NLRP7 overexpression mediates the excessive proliferation of trophoblast cells and suppresses their differentiation, which further enhances their migration and invasion, which ultimately lead to their metastasis to different maternal organs.

## 4. Discussion

The present work demonstrates the mechanism by which NLRP7 contributes to GC growth and aggressiveness. A recent study from our group demonstrated that NLRP7 should not only be considered as the most mutated gene responsible for recurrent CHM, the benign tumor of the placenta that develops into GC in 5–20% of cases but also as an important factor in the etiology of GC, through its direct biological actions on the growth and aggressiveness of GC tumors [2]. This statement is based on strong evidence demonstrated using *in-vitro* and *in-vivo* studies.

First, we demonstrated that NLRP7 overexpression in GC cells drove its function in an inflammasome-independent manner, which conferred to this member of the NOD-like family a new mechanism of action in GC tumor cells. We further demonstrated that the activation of this pathway is associated with the lack of expression of the components of NLRP7 inflammasome machinery, namely ASC, Caspase 1 and IL-1β, in GC cells. Another explanation of this new role of NLRP7 can be related to its closest member, NLRP2, which was found to be highly expressed in JEG-3 compared to HTR8/SVneo cells and its expression was further increased when NLRP7 was knocked down in JEG-3 cells. Importantly, NLRP2 has been reported as a strong inhibitor of NF-κB signaling [29]. Hence, one can speculate that NF-κB decreased activation in JEG-3 cells may cause a decrease in the transcription of Pro-IL-1β, the major effector of the inflammasome function. The increase in NF-κB activation in HTR Sh *NLRP7* compared to JEG Sh *NLRP7* strongly suggests that NLRP7 functions in an inflammasome-independent manner in JEG-3 cells and that NF-κB regulation by NLRP7 is inflammasome-mediated. Another evidence in favor of NLRP7 functioning in an inflammasome-independent manner in these cells emanates from the experiment demonstrating that the specific activator of the NLRP7 inflammasome, FSL-1, only activated NLRP7 inflammasome in HTR8/SVneo but not in JEG-3 cells.

Importantly, NLRP7 is not the unique member of the NOD-like family that may function in an inflammasome-independent manner in tumor settings. NLRP3, another member of the NOD-like family, has also been reported to promote metastasis independently of its inflammasome activity [49]. In addition, targeted deletion of *Nlrp2* in mice lead to lower numbers of lung metastases upon intravenous injection of prostate or melanoma carcinoma cells [50]. Furthermore, a recent study demonstrated that an increased *NLRP12* expression is associated with the progression of prostate cancer in the absence of increased levels of mature IL-1β or IL-18 by cancer cells [51]. Hence, our study strongly suggests that NLRP7, similar to NLRP3 and NLRP12, functions in an inflammasome-independent manner to facilitate cancer progression. Nevertheless, further experiments are warranted to corroborate these findings.

In a recent study from our group, we hypothesized that NLRP7 might contribute to cancer cell camouflage. This conclusion was only based on findings from the immunohistochemistry analyses performed on tumors of placental tissues collected from an in vivo mouse model [2]. Here, we further confirmed, at the cellular level, that *NLRP7* knockdown leads to a decrease in the expression of key genes known to confer camouflage to tumor cells from their environment. These include PD-L1 and the HLA family of proteins as well as the pregnancy hormone hCG. The latter has been reported to be involved in local immune tolerance as it acts as a chemoattractant for T-suppressors (T-Treg) and apoptotic actors for T-lymphocytes [52,53]. Importantly, hCG was also reported to increase the number and activity of regulatory T cells (Treg) [52,53].

The present study also brought evidence of NLRP7 involvement in the differentiation of tumor cells through the induction of embryonic markers such as NANOG, NOTCH1 and OCT3/4, and the activation of their survival through the induction of genes, such as *CD74* and its co-receptor *CXCR2*. These effects are likely to occur in response to NLRP7 overexpression, as they were also observed when the *NLRP7* gene was overexpressed in the non-tumor cells HTR8/SVneo. Hence, one can speculate that these effects are proper to the NLRP7 gene and are totally owing to its overexpression. RNAseq analyses further confirmed NLRP7 involvement in the control of the pathways of cell differentiation, as well as those related to the regulation of immune responses. Altogether, these cellular analyses demonstrate that an increase in the expression of *NLRP7* directly contributes to GC cell survival, camouflage and tumorigenesis.

Further evidence of NLRP7’s direct involvement in GC growth, camouflage, and aggressiveness emanates from the in vivo study demonstrating that *NLRP7* knockdown in JEG-3 cells caused a decrease in their proliferating capacity and an increase in their immunogenicity, which ultimately decreased their overall tumorigenesis.

## 5. Conclusions

In conclusion, our study demonstrates for the first time the mechanism by which NLRP7, independently of its inflammasome, contributes to GC growth and tumorigenesis. Also, this study highlights NLRP7 as a key actor of GC aggressiveness that should be categorized among the important physiopathological factors for the development of this trophoblastic cancer. The clinical relevance of NLRP7 in this rare cancer highlights its potential therapeutic promise as a molecular target to treat resistant gestational choriocarcinoma patients.

## Figures and Tables

**Figure 1 cells-12-00857-f001:**
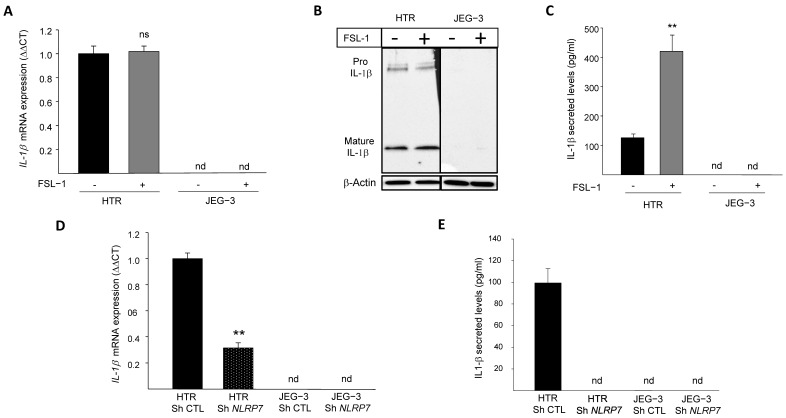
IL-1β responses to FSL-1 stimulation in JEG-3 and HTR8/SVneo cells. (**A**) reports a comparison of the mRNA levels of *IL-1β* expression in HTR8/SVneo and JEG-3 cells that were treated or not by FSL-1 (0.1 µg/mL), a microbial acyl LipoPeptides agonist for NLRP7 (*n* = 3); ns: non-significant, nd: not detected. (**B**) reports a comparison of Pro-IL-1β and mature IL-1β protein expression in JEG-3 and HTR8/SVneo cells cultured in the absence or presence of FSL-1 (0.1 µg/mL). **(C)** reports a comparison of IL-1β secreted levels in JEG-3 and HTR8/SVneo culture media of cells that were treated or not by FSL-1 (0.1 µg/mL) (*n* = 6); ** *p* < 0.01, nd: not detected. (**D**) reports a comparison of the mRNA levels of *IL-1β* expression in HTR Sh CTL, HTR Sh *NLRP7*, JEG-3 Sh CTL and JEG-3 Sh *NLRP7* (*n* = 3); ** *p* < 0.01, nd: not detected. (**E**) reports a comparison of IL-1β secreted levels in culture media of HTR Sh CTL, HTR Sh *NLRP7* and JEG-3 Sh CTL, JEG-3 Sh *NLRP7* (*n* = 3); nd: not detected.

**Figure 2 cells-12-00857-f002:**
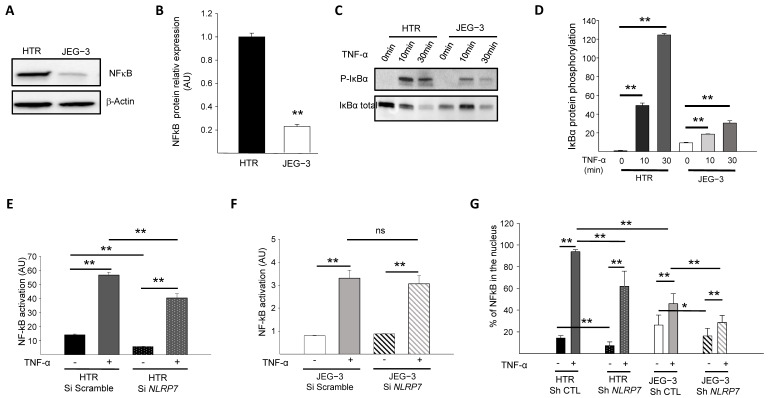
Comparison of the activation of the NF-κB pathway by TNF-α in HTR8/SVneo and JEG-3 cell line silenced for *NLRP7*. (**A**) reports a comparison of NF-κB protein levels in HTR8/SVneo and JEG-3 cells. (**B**) reports a quantification of the NF-κB protein levels in HTR8/SVneo and JEG-3 cells (*n* = 3); ** *p* < 0.01. Standardization of protein signals was performed using antibodies against β-Actin. (**C**) reports a comparison of the phosphorylation of IκBα protein in HTR8/SVneo and JEG-3 cells treated or not with TNF-α (10 ng/mL) for the times indicated. (**D**) reports a quantification of the ratios of phosphorylated IκBα protein (*n* = 3); ** *p* < 0.01. (**E**) reports a comparison of the basal and TNF-α activated NF-κB pathway in HTR Si Scramble and HTR Si *NLRP7* (*n* = 4); ** *p* < 0,01. (**F**) reports a comparison of the basal and TNF-α activated NF-κB pathway in JEG-3 Si scramble and JEG-3 Si *NLRP7* (*n* = 4); ** *p* < 0,01; ns: not significant. (**G**) reports a High Content Analysis (HCA) of the NF-κB nuclear translocation experiment. Results are reported as the percentage of NF-κB present in the nucleus (*n* = 6); * *p* < 0.05, ** *p* < 0.01.

**Figure 3 cells-12-00857-f003:**
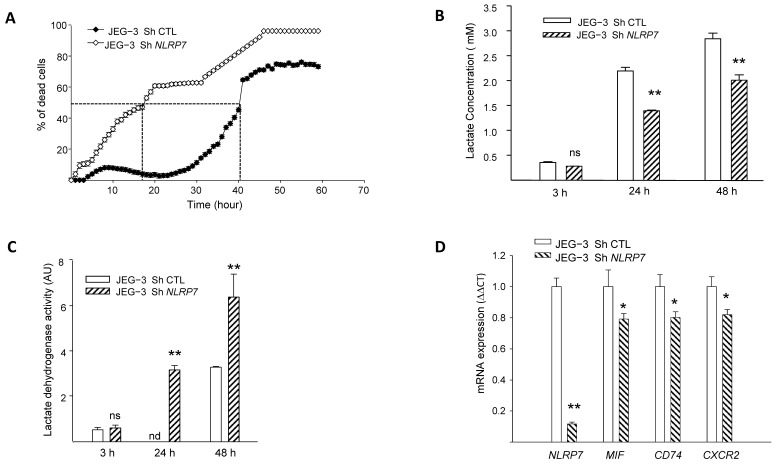
Comparison of JEG-3 cells survival under *NLRP7* knockdown. (**A**) shows the survival curves of JEG-3 Sh CTL and JEG-3 Sh *NLRP7* cells. Cells (10,000 cells/well) were pre-labeled with propidium iodide (5 µg/mL) and then tracked with the Incucyte system, whereby pictures were captured every 2 h (*n* = 12). (**B**) reports a comparison of lactate levels in the culture supernatants of JEG-3 Sh CTL and JEG-3 Sh *NLRP7* cells collected at 3, 24, and 48 h of culture (*n* = 6); ns: not-significant, ** *p* < 0.01. (**C**) reports a comparison of LDH activity in culture supernatants of JEG-3 Sh CTL and JEG-3 Sh *NLRP7* cells collected at 3, 24, and 48 h of culture (*n* = 6); ns not-significant, nd: not detected, ** *p* < 0.01. (**D**) reports comparisons of *NLRP7*, *MIF*, *CD74*, and *CXCR2* mRNA levels in JEG-3 Sh CTL and JEG-3 Sh *NLRP7* cells (*n* = 3); * *p* < 0.05, ** *p* < 0.01.

**Figure 4 cells-12-00857-f004:**
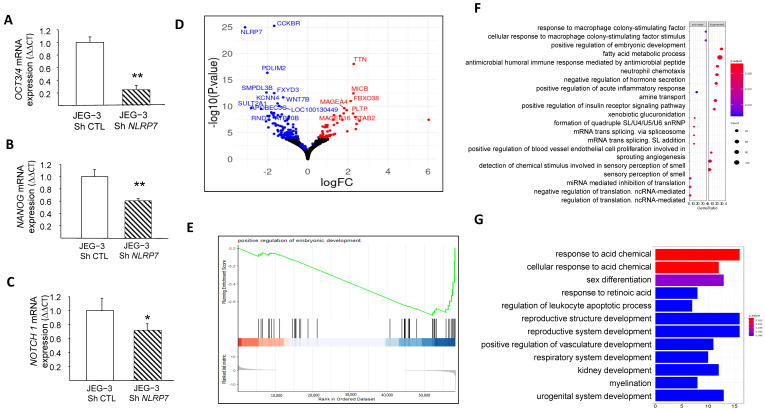
Comparison of JEG-3 cells differentiation under NLRP7 knockdown. (**A**) presents a comparison between JEG-3 Sh CTL and JEG-3 Sh NLRP7 of mRNA levels for *OCT3/4* expression (*n* = 3); ** *p* < 0.01. (**B**) presents a comparison of mRNA levels for *NANOG* expression in JEG-3 Sh CTL and JEG-3 Sh NLRP7 (*n* = 3); ** *p* < 0.01. (**C**) shows a comparison of mRNA levels for *NOTCH 1* expression in JEG-3 Sh CTL and JEG-3 Sh NLRP7 (*n* = 3); * *p* < 0.05. (**D**) displays a volcano plot that presents a logarithmic representation of the fold change between expression levels in JEG-3 Sh *NLRP7* versus JEG-3 Sh CTL for each gene, along with a logarithmic representation of the adjusted *p*-value generated from differential gene expression analysis. Red dots indicate significantly upregulated genes (adjusted *p*-value < 0.05 and absolute log2 fold change > 1.2), and blue dots indicate significantly downregulated genes (adjusted *p*-value < 0.05 and absolute log2 fold change < 1.2). (**E**) reports the GSEA enrichment plot of the regulation of the embryonic development pathway. The running enrichment score (ES) for the gene set is displayed in the top portion of the plot, with a positive ES indicating gene set enrichment at the top of the ranked gene list (i.e., genes overexpressed in JEG-3 Sh *NLRP7*). The middle portion shows the position of the members of the gene set in the ranked list of genes, and the bottom portion displays the logarithmic representation of the fold change between expression levels in JEG-3 Sh *NLRP7* versus JEG-3 Sh CTL cells used as gene ranking metrics. (**F**) presents a dot plot that represents the most significantly enriched molecular processes identified by the GSEA method. The top 20 activated or suppressed pathways were selected based on their enrichment in under- or over-expressed genes in JEG-3 Sh *NLRP7*, respectively, with the gene ratio indicating the proportion of genes deregulated by the pathway in relation to the total number of genes annotated by pathway. The red dashed box highlights key pathways of embryonic development, while the blue dashed box highlights key pathways of adaptive immune responses. (**G**) displays a bar plot that represents the most significantly enriched biological processes identified by the GSEA method. The top 12 activated or suppressed pathways were selected based on their enrichment in under- or over-expressed genes in JEG-3 Sh *NLRP7*, respectively, with the gene ratio indicating the proportion of genes deregulated by the pathway in relation to the total number of genes annotated by pathway. The red dashed box highlights key pathways of embryonic development, while the green dashed box highlights key pathways of adaptive immune responses.

**Figure 5 cells-12-00857-f005:**
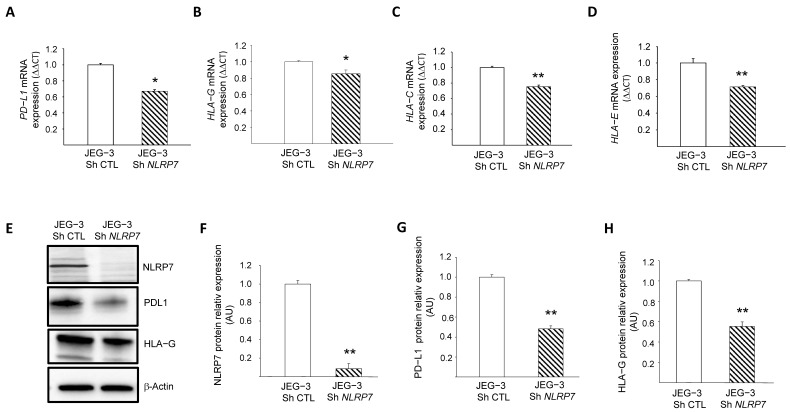
Characterization of the role of NLRP7 in the camouflage of JEG-3 cells. (**A**) reports a comparison of the mRNA levels of *PD-L1* expression in JEG-3 Sh CTL and JEG-3 Sh *NLRP7* (*n* = 3); * *p* < 0.05. (**B**) reports a comparison of the mRNA levels of *HLA-G* expression in JEG-3 Sh CTL and JEG-3 Sh *NLRP7* (*n* = 3); * *p* < 0.05. (**C**) reports a comparison of the mRNA levels of *HLA-C* expression in JEG-3 Sh CTL and JEG-3 Sh *NLRP7* (*n* = 3); ** *p* < 0.01. (**D**) reports a comparison of the mRNA levels of *HLA-E* expression in JEG-3 Sh CTL and JEG-3 Sh *NLRP7* (*n* = 3); ** *p* < 0.01. (**E**) reports a comparison of NLRP7, PD-L1 and HLA-G protein levels in JEG-3 Sh CTL and JEG-3 Sh *NLRP7* cells. (**F**) reports a quantification of the NLRP7 protein levels (*n* = 3); ** *p* < 0.01. Standardization of protein signals was performed using antibodies against β−actin. (**G**) reports a quantification of the PD-L1 protein levels (*n* = 3); ** *p* < 0.01. Standardization of protein signals was performed using antibodies against β-actin. (**H**) reports a quantification of the HLA-G protein levels (*n* = 3); ** *p* < 0.01. Standardization of protein signals was performed using antibodies against β−actin.

**Figure 6 cells-12-00857-f006:**
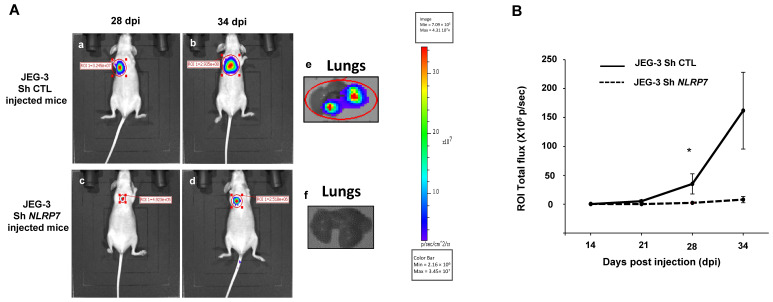
Comparison of tumor growth in mice injected with JEG-3 Sh CTL or JEG-3 Sh *NLRP7* in the tail vein. (**A**) shows representative images of female mice injected with JEG-3 Sh CTL (**Aa**,**b**) *n* = 7 or JEG-3 Sh *NLRP7* (**Ac**,**d**) *n* = 7 at day 28 and 34 post injections (DPI), respectively. Photographs in (**Ae**,**f**) show representative images of lungs collected from JEG-3 Sh CTL or JEG-3 Sh *NLRP7* injected mice, respectively. (**B**) compares tumor lung growth in JEG-3 Sh CTL and in JEG-3 Sh *NLRP7* from 14 to 34 DPI; * *p* < 0.05.

**Figure 7 cells-12-00857-f007:**
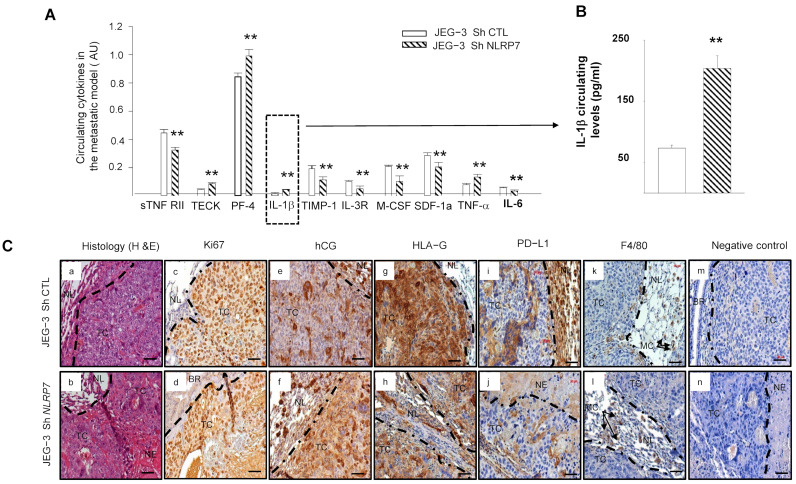
Comparison of the levels of circulating cytokine and immunohistological analyses of the lungs of mice injected with JEG-3 Sh CTL or JEG-3 Sh *NLRP7*. (**A**) reports a selection of circulating cytokines that were reported to be significantly differentially secreted on the panel of the 62 cytokines analyzed by antibody array between JEG-3 Sh CTL (*n* = 7) and JEG-3 Sh *NLRP7* (*n* = 7) tail vein injected mice; ** *p* < 0.01. (**B**) report graphs that compare IL−1β circulating levels in the metastatic animal model. ELISA test was used to compare these levels (*n* = 7 each); ** *p* < 0.01. (**C**) reports representative images of lung sections collected from JEG-3 Sh CTL and JEG-3 Sh *NLRP7* injected mice. Images in (**Ca**,**b**) report the histology of the lungs. The subsequent sections report stainings for the following proteins, Ki67 (**Cc**,**d**); hCG (**Ce**,**f**); HLA-G; (**Cg**,**h**); PDL-1 (**Ci**,**j**) and F4/80 (**Ck**,**l**). Images in (**Cm**,**n**) represent negative controls that were incubated without the primary antibodies. TC: tumor cells, NL: Normal Lung, BR: Bronchus, MC: Macrophages, NE: Necrosis. Scale bar = 50 µm.

**Figure 8 cells-12-00857-f008:**
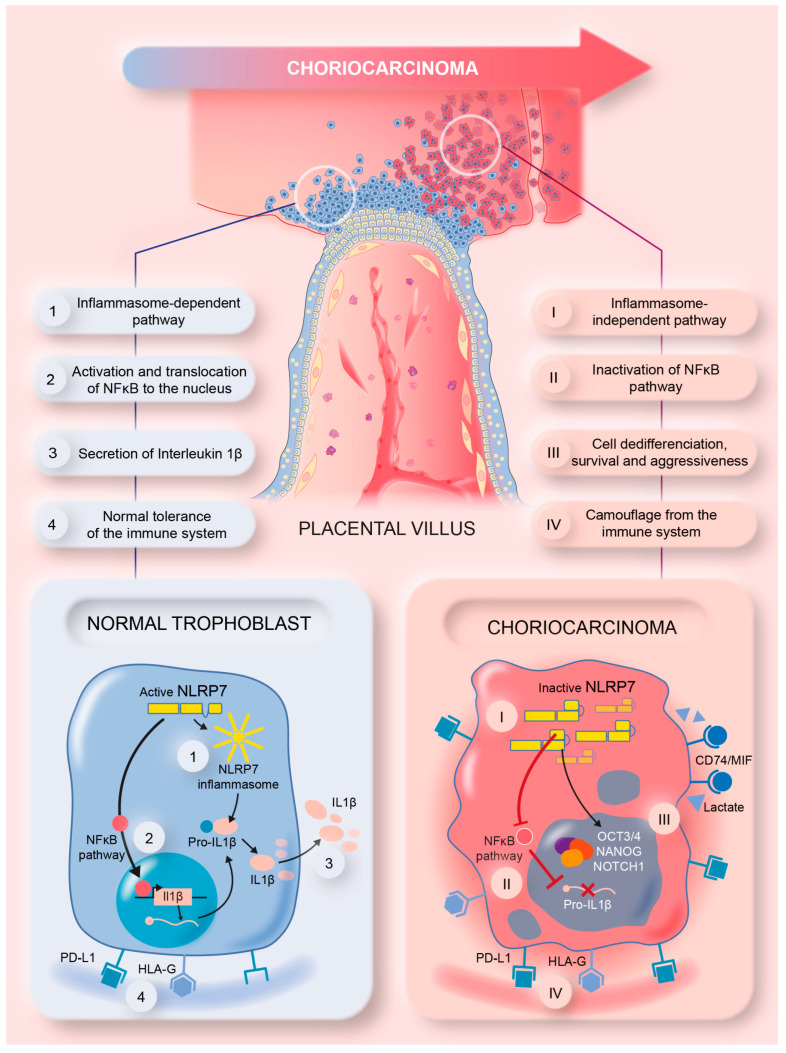
A summary model illustrating the mechanism by which NLRP7 contributes to choriocarcinoma tumorigenesis.

**Table 1 cells-12-00857-t001:** Primers used for real-time RT-PCR.

Gene	Forward primer (5′–3′)	Reverse primer (5′–3′)	Temperature (°C)
*NLRP7*	TGCTGTACAAGACCATGACACG	ACTCAAGCCCTCACACAGAAAC	60
*NLRP2*	TGGATCAAATAAGGATCTGATGG	AGCTAGGCAGAGGTTCCGATG	60
*CASP1*	TGCCTGTTCCTGTGATGTGG	TGTCCTGGGAAGAGGTAGAAACATC	60
*IL-1β*	GTCGGAGATTCGTAGCTGGAT	GTCGGAGATTCGTAGCTGGAT	60
*OCT3/4*	CCTGAAGCAGAAGAGGATCACC	AAAGCGGCAGATGGTCGTTTGG	60
*NANOG*	TTGGGACTGGTGGAAGAATC	GATTTGTGGGCCTGAAGAAA	60
*NOTCH1*	GGTGAACTGCTCTGAGGAGATC	GGATTGCAGTCGTCCACGTTGA	60
*PD−L1*	CAGTTCTGCGCAGCTTCC	TTCAGCAAATGCCAGTAGGTC	60
*HLA−G*	GCTGCCCTGTGTGGGGACTGAGTG	ACGGAGACATCCCAGCCCCTTT	60
*HLA−C*	GTGTCCACCGTGACCCCTGTC	ATTCACGTTCTTAACTTCAT	60
*HLA−E*	GCACACATTTTCCGAGTGAAT	CAGCCATGCATCCACTGC	60
*MIF*	CCGGACAGGGTCTACATCA	ATTTCTCCCCACCAGAAGGT	60
*CD74*	GACCTTATCTCCAACAATGAGCAAC	AGCAGAGTCACCAGGATGGAA	60
*CXCR2*	ACATGGGCAACAATACAGCA	GAGGACGACAGCAAAGATG	60
*18S*	AAACGGCTACCACATCCAAG	CCTCCAATGGATCCTCGTTA	60

## Data Availability

All data in our study will be available upon reasonable request.

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
