# Peer review of "NLRP7 Enhances Choriocarcinoma Cell Survival and Camouflage in an Inflammasome Independent Pathway"

_cells, 2023, doi:10.3390/cells12060857_

Round 1
Reviewer 1 Report
In this study, the authors investigated the potential mechanisms of NLRP7 in GC growth and tumorigenesis of gestational choriocarcinoma in vitro and validated the results in vivo. It is an interesting work. However, there are several minor issues need to be demonstrated.
1. The authors used female mice in the animal study. It is well-known that hormone levels are different during the cycle. Did the cycle affect the metastasis of the choriocarcinoma cells?
2. In the Figure 6Ae and f, is there any difference between organs from the two groups? If no, remove these images. In addition, resolution of the organ images is too low.
3. Define the following words for the first presence in the text.
Line 49: GTD
Line 57: NLRP7
4. Line 58-59: delete (Nucleotide binding oligomerization domain, Leucine-Rich repeat family, 58 Pyrin domain containing 7).
Author Response
Responses to Reviewer 1
Comments and Suggestions for Authors
In this study, the authors investigated the potential mechanisms of NLRP7 in GC growth and tumorigenesis of gestational choriocarcinoma in vitro and validated the results in vivo. It is an interesting work. However, there are several minor issues need to be demonstrated.
- The authors used female mice in the animal study. It is well-known that hormone levels are different during the cycle. Did the cycle affect the metastasis of the choriocarcinoma cells?
Response1: We thank the reviewer for this relevant comment. We agree that hormones, especially sex hormones may affect cancer metastasis. Regarding gestational choriocarcinoma, and as in any pregnancy, females are not cycling, so no study has addressed the potential impact of sex hormones on this type of cancer. The use of female mice in the present study was chosen by the fact that gestational choriocarcinoma can only develop in female gender. Because our mice were not gravid, a potential cycling could occur differently between mice. To avoid this potential effect, we conducted our studies on females that have been housed together for, at least 2 weeks, before enrolment into the study. This grouped housing has been reported to synchronize the menstrual cycle in the cohort (W K Whitten 1959).To overcome this potential bias, housed mice (x4 per cage) were enrolled blindly into control (x2 mice) or treated (x 2 mice) until reaching the required number of mice used in each group. This information is now added to the Material and Methods section of the revised manuscript.
- In the Figure 6Ae and f, is there any difference between organs from the two groups? If no, remove these images. In addition, resolution of the organ images is too low.
Response 2:The images on Figure 6Ae and Figure 6Af report representative images of collected organs that were imaged following culling of the mice. We agree with the reviewer that the differences are to be considered for the changes observed between the lungs in the two groups. As suggested by this reviewer, in the revised manuscript we have incorporated the images pertaining specifically to the lungs of the mice injected either with JEG3-Sh CTL and/or with JEG-3-Sh NLRP7 and illustrated those images in Figure 6Ae and Figure 6Af, respectively
- Define the following words for the first presence in the text.
Line 49: GTD
Response3: We thank the reviewer for this important remark. The abbreviation GTD is now defined in page2-line 45. GTD stands for Gestational Trophoblastic Disease.
Line 57: NLRP7
Response 4: It is important to accurately reference and locate technical terms and their definitions in the text. The definition of the word NLRP7 has now been changed to define its first citation. See page2 line 54.
- Line 58-59: delete (Nucleotide binding oligomerization domain, Leucine-Rich repeat family, Pyrin domain containing 7).
Response 5:The name "Nucleotide binding oligomerization domain, Leucine-Rich repeat family, Pyrin domain containing 7" has been relocated to line 57 in the manuscript.
Reviewer 2 Report
In this study the authors examined a possible pro-tumorigenic function of NLRP7 in choriocarcinoma cell, and showed its inflammasome-independent pathway. The results are partly interesting, but several points should be clarified.
Comments.
1, Strong induction of cell death with sh-NLRP7 (Fig.3A) appears quite impressive, and suggests a direct anti-apoptotic activity of NLRP7 in choriocarcinoma. Does this cell death also occur in 3D-culture condition of JEG3-shNLRP7 cell ?
2, They used only one cell line, JEG3. Is this anti-cell death activity of NLRP7 also observed in other choriocarcinoma cell lines such as BeWo ?
3, In the title of this paper, the authors suggest that NLRP7 gene controls choriocarcinoma cell differentiation, but the results appear to suggest that the NLRP7 gene suppression using shRNA directly induce cell death, not cell differentiation. Please discuss a little more on this point.
4, IL-1beta level: in Fig. 1E, JEG3-sh-control and JEG3-shNLRP7 expressed undetectable levels of IL-1beta. However, in Fig. 7, circulating IL-1beta in tumor cell-injected mice were increased in the mice with JEG3-shNLRP7 compared to those with JEG3-sh-cont. Is this IL-1beta in the blood the JEG3-derived human IL-1beta ? If so, how do the authors explain a difference between the results in Fig.1 and Fig.7 ?
5, In Fig. 8: In both cells (normal cell and choriocarcinoma), the authors indicate “inactive NLRP7”. However, in choriocarcinoma cells, does this inactive form of NLRP7 really work for the blocking /suppression of NF-kB and OCT3/4 activation pathway ? Or a different active form of NLRP7 protein works for the modification of these pathways ?
6, Fig. 4 F,G: words appear too small to see.
Author Response
Plesae find the response to reviewer attached as PDF file.

Reviewer 3 Report
In this manuscript, the authors evaluated the role of NLRP7 in the regulation of diverse caracteristics in both non-tumoral (trophoblasts) and tumoral cells (choriocarcinoma cell line). Using different constructs and methods, they either overexpressed or knocked out NLRP7 in both cell lines and evaluated its role in the regulation of cellular mecanisms such as proliferation and survival. Their results showed that the effect of NLRP7 in tumoral cells act through an inflammasome-independant pathway (Figure 1 and S1), caracterized by a decreased NF-kB activity in tumoral cells. The loss of NLRP7 in tumoral cells resulted in increased cell death, whereas other roles elucidated include the dedifferentiation of tumoral cells and their potential camouflage/evasion of the immune system. These roles were determined through in vitro study of target genes implicated in both mecanisms. These findings were then confirmed through an in vivo model of tumor growth in mice injected with either JEG-3 Sh CTRL or JEG-3 Sh NLRP7. The latter showed decreased tumor size and increased host immune mediated response against the cells.
Upon further reading, I recognize the immense work invested in this manuscript. The presentation and the scientific value are noticeable. The authors described with great details the material and method and used numerous tools to support their hypothesis, including the use of both in vitro and in vivo models. The results and discussions further address the potential questions that would arise.
Once again, I would like to recognize the quality of this manuscript in its actual form.
Author Response
Reviewer 3
Comments and Suggestions for Authors
In this manuscript, the authors evaluated the role of NLRP7 in the regulation of diverse caracteristics in both non-tumoral (trophoblasts) and tumoral cells (choriocarcinoma cell line). Using different constructs and methods, they either overexpressed or knocked out NLRP7 in both cell lines and evaluated its role in the regulation of cellular mechanisms such as proliferation and survival. Their results showed that the effect of NLRP7 in tumoral cells act through an inflammasome-independant pathway (Figure 1 and S1), characterized by a decreased NF-kB activity in tumoral cells. The loss of NLRP7 in tumoral cells resulted in increased cell death, whereas other roles elucidated include the dedifferentiation of tumoral cells and their potential camouflage/evasion of the immune system. These roles were determined through in vitro study of target genes implicated in both mecanisms. These findings were then confirmed through an in vivo model of tumor growth in mice injected with either JEG-3 Sh CTRL or JEG-3 Sh NLRP7. The latter showed decreased tumor size and increased host immune mediated response against the cells.
Upon further reading, I recognize the immense work invested in this manuscript. The presentation and the scientific value are noticeable. The authors described with great details the material and method and used numerous tools to support their hypothesis, including the use of both in vitro and in vivo models. The results and discussions further address the potential questions that would arise.
Once again, I would like to recognize the quality of this manuscript in its actual form.
Response: We would like to thank the reviewer for their recognition of the merits of the present work. Authors of the manuscript appreciate the reviewers’ positive and constructive feedback to further improve the quality of the revised manuscript. Furthermore, as highlighted by the reviewer, this study has several merits, including the use of multiple tools to support the hypothesis, as well as the comprehensive approaches using both in vitro and in vivo models. We strongly believe that this study will advance our understanding of the mechanism(s) by which NLRP7, one of the most mutated gene in CHM, play a significant role in the progression of tumor cell pathogenesis in gestational choriocarcinoma.
Reviewer 4 Report
The study clearly demonstrates the functional role of NLRP7 inflammasomes in cell differentiation and camouflage independent of the inflammasome pathway. The study is well-designed, and the authors use biochemical and genetic approaches to verify the role of NLRP7 inflammasomes. The authors should address the following points:
1) In line no. 131, Puromycin was used at a concentration of 200 ug/ml, which seems to be a very high concentration.
2) The authors have demonstrated the silencing effect of NLRP7 in HTR and JEG-3 cells. How do the authors measure the silencing effect of NLRP7 in these cell lines?
3) In supplementary Fig S2 B, HTR cells overexpressing NLRP7 showed a robust increase in C-myc expression, a critical transcription factor involved in regulating various cell survival genes. The authors should describe the c-myc upregulation in the discussion section. Does NLRP7 overexpression have any effect on N-myc expression?
4) The authors use JEG-3 cell line only to compare with normal cells (HTR8/SVneo) and make conclusions based on the obtained data. Do the authors validate the data in other choriocarcinoma cell lines to confirm the functional role of NLRP7?
Author Response
Plesae find responses to reviewer4 attached bellow

Round 2
Reviewer 2 Report
The authors answered to the comments, and the manuscript was well revised. The reviewer has no additional comments.
Reviewer 4 Report
I am thankful to the authors for completely justifying the raised comments. The raised issues and comments are fully provided with sufficient evidence.